# Exploration of Factors Predicting Sport Coaches’ Perceived Performance

**DOI:** 10.3390/sports13030083

**Published:** 2025-03-10

**Authors:** Kathrine Lervold, Jan Arvid Haugan, Maja Gunhild Olsen Østerås, Frode Moen

**Affiliations:** 1Department of Teacher Education, Faculty of Social and Educational Sciences, Norwegian University of Science and Technology, 7491 Trondheim, Norway; kathrine.lervold@ntnu.no; 2Department of Education and Lifelong Learning, Faculty of Social and Educational Sciences, Norwegian University of Science and Technology, 7491 Trondheim, Norway; maja.olsen@ntnu.no (M.G.O.Ø.); frode.moen@ntnu.no (F.M.); 3Centre for Elite Sports Research, Department of Neuromedicine and Movement Science, Faculty of Medicine and Health Sciences, Norwegian University of Science and Technology, 7491 Trondheim, Norway

**Keywords:** coach performance, predictive factors, mentoring, coach–athlete development, professional development

## Abstract

This study explores the predictive factors influencing sport coaches’ perceived performance levels following an 18-month mentor-based education program. The study employed a quasi-experimental, pre-test/post-test control group design to assess changes in perceived coach performance over time. The experimental group participated in six group gatherings and at least 15 individual mentoring sessions over 18 months, while the control group received no intervention. Participants were recruited from a non-formal coach education program run by the Norwegian Olympic Sports Center (NOSC), requiring recommendations from their respective sport federations. Of the 159 coaches who applied, 73 were selected for the program and invited to participate, along with 29 additional coaches from specialized high schools for elite sports, forming a total sample of 98 coaches (69 in the experimental group, 29 in the control group) at pre-test. The sample consisted of 61 males (62%) and 37 females (38%), aged 26 to 71 years (M = 38.3, SD = 8.3), representing over 20 sports, with handball (15.3%), cross-country skiing (10.2%), soccer (7.1%), and track and field (7.1%) being the most common. Data collection included an online questionnaire measuring perceived coach performance, coaching hours, age, and dimensions of the coach–athlete working alliance (task, bond, and goal development). After 18 months, 75 coaches completed the study, yielding a response rate of 73.5%. Hierarchical regression analyses revealed that coaches’ ages, weekly coaching hours, baseline perceived performance, and task development within the coach–athlete working alliance positively predicted their performance perception post-test. Participation in the mentor-based program also had a significant positive effect. However, neither bond nor goal development in the working alliance predicted performance perception. These findings underscore the importance of mentoring, structured self-reflection, and task-focused coaching strategies in enhancing coaching effectiveness. The results have implications for coach education programs seeking to foster professional growth and performance development.

## 1. Introduction

Sport coaches’ perceptions of their own performance are crucial in understanding their effectiveness and the overall success of their coaching practices [1,2]. Research indicates that coaches’ self-assessment of their effectiveness often aligns with their level of job satisfaction and the perceived outcomes of their coaching [3,4]. Coaches that perceive their own performance level to be high tend to set higher goals and exhibit more positive coaching behaviors, which can lead to better athletic performance and satisfaction [5].

Self-perception is influenced by several factors, including feedback from athletes, colleagues, and the sport coaches own reflective practices [6,7]. For example, coaches who engage in regular reflective practices tend to have a more accurate self-assessment of their strengths and areas for improvement [8]. In addition, age and experience are pivotal in shaping coaching effectiveness and approaches. Older coaches often bring a wealth of experience and a broad understanding of the game, which can be advantageous [9,10]. However, age can also impact coaching styles and adaptability. Research suggests that older coaches may rely more on traditional methods and may be less inclined to adopt new coaching technologies or strategies [11,12]. On the other hand, experience, regardless of age, has been shown to positively correlate with coaching efficacy [13,14]. Experienced coaches generally have refined their coaching techniques and developed a deeper understanding of athlete development and team dynamics [15]. Experience allows coaches to better handle the complexities of coaching, such as managing diverse personalities and resolving conflicts [16,17].

Furthermore, the amount of time coaches dedicate to their role each week can significantly influence the coaches’ perceptions of their own performance levels. Studies indicate that more hours spent on coaching activities often lead to greater skill development and more successful coaching outcomes [18,19]. However, the quality of coaching time is equally important. For example, it is not just about the quantity but the effectiveness of time spent in planning, training, and interacting with athletes [20]. Coaches who spend more time engaging in direct coaching activities, such as training sessions and one-on-one interactions, tend to develop stronger relationships with their athletes and have a more profound impact on their performance [21]. Conversely, excessive time spent on coaching without adequate balance can lead to burnout and decreased effectiveness [22].

The coach–athlete relationship is a critical factor in determining coaching effectiveness and athlete satisfaction [23,24]. A positive coach–athlete relationship is characterized by trust, respect, and open communication, which enhances athletes’ motivation and performance [12,25]. Research shows that strong relational bonds between coaches and athletes contribute to better team cohesion, improved performance, and higher levels of athlete and coach well-being [26]. The quality of the coach–athlete relationship is influenced by the coach’s interpersonal skills, empathy, and the ability to provide constructive feedback. Coaches who invest in building strong, supportive relationships with their athletes are more likely to create an environment conducive to high performance and personal growth [27]. One way to foster sport coaches’ skills, abilities, knowledge and attitudes that establish, develop, repair and maintain relationships between athletes and coaches are mentor-based coach education programs.

Coach education programs that incorporate mentoring have been shown to be effective in enhancing coaching skills and practices [28,29]. Mentoring provides coaches with guidance, support, and feedback from more experienced professionals, which can lead to significant improvements in coaching effectiveness [30,31]. Research highlights that mentoring helps coaches develop both technical skills and soft skills, such as communication and leadership, which are crucial for successful coaching [32]. Programs that use mentoring also facilitate the development of a reflective practice among coaches, encouraging them to evaluate and improve their own coaching methods [33]. These programs often include structured interactions, regular feedback, and opportunities for collaborative learning, which contribute to professional growth and improved coaching practices [34].

In summary, the connections between coaches’ self-perception of their performance level, age, time spent coaching, the coach–athlete relationship, and the impact of mentoring-based coach education programs are intricate and interrelated. Coaches’ self-perception affects their motivation and effectiveness, while age and experience influence their adaptability and approach. The amount of time spent on coaching activities can impact effectiveness, but balance is crucial to avoid burning out. The quality of the coach–athlete relationship is a significant predictor of success, and mentoring programs play a vital role in enhancing coaching practices and professional development. Understanding these connections can provide valuable insights for improving coaching strategies and fostering better outcomes for athletes.

### The Current Study

The aim of the current study was to investigate predictive factors of perceived coach performance over a period of 18 months following a mentor-based coach education program that was held among elite coaches. While research has extensively examined factors influencing sport coaches’ self-perceived performance, there is a lack of studies investigating whether participation in structured, long-term mentor-based education programs predicts changes in perceived coaching effectiveness. Previous research on coach development has focused on informal learning (e.g., experience, self-reflection, and peer learning) rather than structured mentorship interventions. This study addresses this gap by examining how participation in 18-month non-formal coach education program influences coaches’ perceptions of their performance over time, thereby providing empirical evidence for the role of structured mentorship in professional development. Unlike previous research, which often relies on cross-sectional or qualitative designs to examine coaching effectiveness, this study utilizes a pre-test/post-test control group design to assess the direct impact of mentorship-based education programs on perceived coach performance. This approach allows for a more rigorous assessment of the causal relationships between structured mentorship, working alliances, and self-perceived performance, contributing valuable empirical evidence to the field of coach development. The following hypothesis were developed in the current study: a coach’s perception of their own performance is positively predicted by:-H1: The coaches’ ages [13,14];-H2: Coaching hours per week [18,19];-H3: Their perception of their performance level at pre-test [3,4];-H4: The coach–athlete relationship [20,27];-H5: The participation in the mentor-based education program [33,34];

## 2. Method

### 2.1. Study Design

This study employed a quasi-experimental, pre-test/post-test control group design to assess predictive factors influencing sport coaches’ perceived performance. The experimental group participated in an 18-month mentor-based education program, while the control group received no intervention. The 18-month mentor-based education program was implemented by the NOSC and consisted of six group gatherings, each focusing on key coaching themes such as leadership, athlete development, and reflective practice. The coaches’ that participated received a minimum of 15 individual mentoring sessions, where coaches received personalized feedback and guidance from experienced mentors through self-reflection activities, structured discussions, and practical application of coaching strategies. Mentors, selected by the NOSC, all had elite coaching experience and were required to complete a university-accredited mentoring education program (7.5 ECTS credits) at the Norwegian University of Science and Technology (NTNU). The control group did not receive any structured mentoring or educational intervention but continued their normal coaching activities. Data were collected at two time points (pre-test and post-test) through online questionnaires, measuring perceived coach performance, coaching hours, age and dimensions of the coach–athlete working alliance (task, bond and goal development). The study used hierarchical regression analysis tests to assess changes over time and identify significant predictors of perceived coach performance.

### 2.2. Participants and Recruitment

Participants were recruited from a non-formal coach education program administered by the Norwegian Olympic Sport Center (NOSC). Admission to the program required a formal recommendation from national sport federations, ensuring that selected coaches were working with talented junior athletes aspiring to reach elite levels.

Of the 159 coaches who applied, 73 were selected based on their qualifications, coaching experience, and the endorsement of their federations. These 73 coaches were invited to participate in the study as the experimental group. Additionally, 29 elite coaches from high schools specializing in elite sports coaching were recruited as the control group, forming a total pre-test sample of 98 coaches (69 in the experimental group, 29 in the control group). Due to the nature of the intervention, random assignment was not feasible, making this a quasi-experimental study. The control group did not receive any structured mentoring but continued their regular coaching activities. As participants were selected based on recommendations from their sport federations, those chosen for the program may have had greater motivation or prior development, which could influence their performance perceptions.

After 18 months, 75 of the 98 coaches completed the post-test survey, resulting in a response rate of 73.5%. Of these, 54 were from the experimental group (78%) and 21 were from the control group (72%). The final sample consisted of 61 males (62%) and 37 females (38%), aged 26 to 71 years (M = 38.3, SD = 8.3), representing over 20 sports. The most common sports included handball (15.3%), cross-country skiing (10.2%), soccer (7.1%), and track and field (7.1%). Coaches in the experimental group worked an average of 36 h (SD = 15) per week, while those in the control group worked an average of 20 h (SD = 13) per week.

### 2.3. Instruments

To investigate predictive factors of perceived coach performance over a period of 18 months following a non-formal coach education program, variables measuring their age, hours spent in coaching practices, the working alliances between the coaches and their athletes, and a group variable were needed. To measure these variables, an online questionnaire was developed that detected descriptive information about the coaches’ work situation (their sport, their athletes’ performance level, their role as coach, the time they spend on coaching) and their experiences related to their work as coaches. All measurements were based on previously developed scales proven to hold both satisfactory validity and reliability. The questionnaire included scales to measure the coaches’ perceived coach performance and the working alliance between the coach and their athletes. All measurements were taken in Norwegian, and the questionnaire was conducted online and took approximately 15–20 min to complete. The measurements are described below in more detail.

#### 2.3.1. The Coach–Athlete Working Alliance (CAWA)

The coaches’ perception of the coach–athlete working alliance was measured by the coach–athlete working alliance (CAWA) [35].The scale is an adjusted version of the working alliance inventory [36].The scale consists of three subscales measuring the degree of agreement concerning goals related to the coach–athlete working alliance, the tasks chosen to achieve these goals, and the personal bond between the coach and the athlete. Each subscale has 4 items associated with goals (e.g., “The coach and athlete are working on mutually agreed upon goals”), tasks (e.g., “There is agreement about the steps taken to help improve the athlete’s situation”), and bonds (e.g., “There is mutual trust between the coach and athlete”). The coaches were asked to respond on a 7-point scale ranging from (1) never to (7) always, indicating to what degree the statement applied to them and their coach–athlete relationships. The scale can be used as a three-, two-, and one-dimensional scale. For the current study, a one-dimensional scale was chosen. Cronbach’s alpha for each subscale at pre/post was 0.61/0.69 (goal), 0.70/0.70 (task), 0.70/0.79 (bond), and, for the complete measurement, 0.83.

#### 2.3.2. Perceived Coach Performance (PCP)

The perceived coach performance (PCP) scale measures coaches’ perceived satisfaction with progress related to their task performances as coaches. The scale is an adjusted version of the Athlete Satisfaction Questionnaire [37]. Coach performance includes coaches’ perception of absolute performance, improvements in performance, and goal achievement. An example of an item is “I am satisfied with the degree I have reached my performance goals during the season”. The coaches were asked to consider 4 items and how satisfied they were with their own progress as coaches in their sports during the last year on a 7-point scale ranging from 1 (not at all satisfied) to 7 (extremely satisfied). The Cronbach’s alphas for the scale at pre/post were 88/83.

#### 2.3.3. Additonal Variables

In addition to the mentioned instruments, we included variables that measured demographics and coaching background (age, gender, sport, coaching role, weekly coaching hours and employment status) and a group variable (a binary variable (0 = Control, 1 = Experiment) indicating whether the coach participated in the mentor-based education program.

### 2.4. Statistical Analyses

Composite scores for each of the included questionnaires and their respective subscales were calculated according to their relevant scoring manuals. Then, preliminary analyses were conducted to ensure no violation of the assumptions of normality, linearity, multicollinearity, and homoscedasticity. Descriptive statistics, such as statistical means, standard deviations, and maximum and minimum values, were analyzed for the pre-test and the post-test. To test the hypothesis that the coach program improves the coach–athlete working alliance, the perceived coach performance, a coach’s perception of self, and social resources in a sport, a data analysis procedure with these variables was conducted. Firstly, descriptive statistics including statistical means and standard deviations measuring the investigated variables were carried out at each testing time point.

To investigate whether the coaches’ ages, the number of hours the coaches spent on their coaching pr week, PCP at pre-test, the different dimensions of the CAWA (bond, goal, task), and the group variable predicted their PCP at post test, a series of four steps of linear regression analyses were conducted. The age of the coach, the amount of time coaches spent on their work as coaches pr week, the PCP pre-test score, the CAWA dimensions, and the group variable were entered as independent variables, and the PCP post-test score was included as a dependent variable.

## 3. Results

This section presents the findings from the study, examining the predictive factors influencing sport coaches’ perceived performance following an 18-month mentor-based education program. The results are organized into three key areas: descriptive analyses, correlation analyses, and hierarchical analyses. First, descriptive statistics provide an overview of the sample characteristics and distributions of key study variables at both pre-test and post-test. Second, correlation analyses explore the relationships between the core study variables, identifying significant associations between coach–athlete working alliance dimensions and perceived coach performance. Finally, the hierarchical regression analyses assess the predictive power of various factors, including age, weekly coaching hours, baseline perceived performance, coach–athlete working alliance dimensions, and participation in the mentorship program, on post-test perceived performance. The findings offer insights into the role of structured mentorship and key coaching factors in shaping coaches’ self-perceived effectiveness over time.

### 3.1. Descriptive Analyses

Descriptive statistics of coaches’ scores on the coach–athlete working alliance variables and perceived coach performance from the pre-test and post-test are in Table 1 and Table 2, respectively. Table 1 presents the means, standard deviations, and Cronbach’s alpha coefficients for all variables at pre-test.

Table 2 presents the same statistics from post-test alongside Pearson correlation coefficients among the study variables.

### 3.2. Correlation Analyses and Hierarchical Regression Analyses

Pearson correlations between variables taken during at pre-test and post-test are included in Table 1 and Table 2. At post-test, stronger positive correlations were observed between the CAWA-task and PCP (r = 0.40, *p* < 0.05). However, trends suggest an increase in scores for the experimental group (PCP: 4.83 → 4.90), while scores in the control group decreased (PCP: 4.76 → 4.56).

The stepwise hierarchical linear regression analyses revealed the significant effects of age, coaching hours per week, PCP at pre-test, CAWAI-task-development, and the group variable at the last step on the PCP at post-test. The results of the four steps of the regression analyses are summarized in Table 3.

The stepwise hierarchical linear regression analyses in Table 3 revealed the significant effects of age, coaching hours per week, PCP at pre-test, CAWAI-task, and the group variable at the last step on the PCP at post-test. The strongest predictors are PCP at pre-test together with age and the group variable. It is worth noting that the correlation analyses in Table 1 and Table 2 revealed high correlations between the CAWAI dimensions, and such interrelated associations interrupt the variables’ predictive power on the dependent variable. Thus, the results show the CAWAI-task dimension’s unique relationship with PCP at post-test. The predictor variables uniquely explain 25% of the variance in the PCP at post-test in the last step.

## 4. Discussion

The purpose of this study was to examine factors predicting sport coaches’ perceived performance following an 18-month mentor-based education program. The results indicate that age, coaching hours per week, initial perceived performance at pre-test, the task-dimension within the coach–athlete working alliance, and participation in the mentor-based program positively predicted perceived performance at post-test. However, neither bond nor goal development within the working alliance significantly predicted perceived performance. One key result is the modest predictive power of the regression model (R^2^ = 0.25), suggesting that additional unmeasured factors contribute to coaches’ self-perceived performance. This highlights the complexity of coaching effectiveness, where variables such as individual coaching philosophy, resilience, and athlete feedback may play a crucial role. These findings have implications for both theory and practice and will be discussed in the following sections.

### 4.1. The Role of Age and Coaching Hours in Perceived Performance

Consistent with previous research, age positively predicted perceived performance, suggesting that experience accumulated over time enhances self-perception [13,14]. This can be explained through expertise development theory [38], which posits that deliberate practice over time leads to improved competence. Older coaches often have more exposure to complex coaching situations, diverse athlete personalities, and long-term athlete development, which may contribute to greater confidence in their coaching abilities.

However, experience alone does not guarantee improvement. Research suggests that engagement in reflective practices and continuous learning is essential for coaches to refine their skills [4] Future research should investigate whether coaching philosophies and adaptive learning strategies moderate the relationship between age and perceived performance.

Similarly, weekly coaching hours positively predicted perceived performance. This aligns with the notion that increased time spent on coaching allows for more learning opportunities, skill refinement, and feedback [18,19]. However, while more time invested can be beneficial, it is also associated with a higher risk of burnout [22]. Future studies should examine how workload balance and self-care strategies impact long-term coaching effectiveness.

### 4.2. The Importance of Initial Performance Perception

Perceived performance at pre-test was one of the strongest predictors of perceived performance at post-test, reinforcing self-efficacy theory [39]. According to Bandura’s model, individuals who believe they are competent in a domain are more likely to engage in mastery-oriented behaviors, persist through challenges, and perceive future success [40].

This finding has important implications for coach education programs. Coaches who start with low self-perceived competence may require additional support, confidence-building exercises, and structured feedback to develop a stronger sense of self-efficacy. Future research should explore whether interventions tailored to coaches with low initial self-perceptions lead to greater improvements in perceived performance.

### 4.3. Task Development as a Predictive Factor of Perceived Performance

A key finding was that the task dimension within the coach–athlete working alliance significantly predicted perceived performance, whereas the bond and goal dimensions did not. The task dimension refers to the practical, skill-based aspects of coaching, such as planning training sessions, setting short-term goals, and providing feedback. This aligns with the achievement goal theory [41] which emphasizes that task-oriented coaching strategies foster intrinsic motivation and skill mastery. Coaches who perceive that their perceptions of important tasks are aligned with the athletes’ perceptions are likely to perceive themselves as effective and competent [42].

While bond and are essential for relationship quality, they did not predict self-perceived coaching performance. This finding suggests that perceived effectiveness is more strongly tied to observable, performance-driven improvements rather that emotional connections or long-term goal setting. One possible explanation is that high-performance coaching environments prioritize immediate, task-driven actions over relational aspects [43]. While a strong coach–athlete bond can enhance athlete motivation, it may be less directly linked to how coaches evaluate their own performance. Similarly, goal development, which involves long-term aspirations, may not provide the same immediate feedback mechanisms that contribute to self-perception.

Future research should examine whether bond and goal development predict other coaching outcomes, such as athletes’ satisfaction, motivation, or retention.

### 4.4. The Impact of the Mentor-Based Education Program

Participation in the mentor-based education program significantly predicted perceived performance, supporting research on the benefits of mentorship in coach development [28,29]. Mentorship is widely recognized as a powerful tool for professional development. The program likely provided structured opportunities for reflection, skill acquisition, and knowledge-sharing, which may explain its association with perceived performance. However, the predictive value of the regression model (R^2^ = 0.25) suggests that many other factors contribute to the coaches’ self-perceptions.

Future studies should directly compare intervention and control groups to determine whether mentor-based programs lead to significant improvements compared to other forms of coach education.

### 4.5. Limitations with the Study

This study has several limitations that should be acknowledged.

First, the quasi-experimental design prevents strong causal inferences. The coaches were not randomly assigned to the experimental or control group and no direct between-group comparisons were made. Future research should implement randomized controlled trials (RCTs) to strengthen causal claims.

Second, the regression model’s predictive power was relatively low (R^2^ = 0.25). This suggests that many unmeasured factors contribute to perceived performance, such as coaching philosophy, psychological resilience, and athlete feedback. Future research should incorporate qualitative methods to capture a more nuanced understanding of coaches’ experiences.

Third, the study relied exclusively on self-reported measures of performance. While self-perception is a valuable indicator of confidence and motivation, it does not necessarily reflect actual coaching effectiveness. Future studies should include athlete performance data, third-party evaluations, or video analyses to provide a more comprehensive assessment.

Finally, a limitation of this study is that no a priori sample size calculation was conducted to determine the required number of participants for adequate statistical power. Consequently, the study’s findings should be interpreted with caution, as the sample size may have influenced the strength of the detected effects.

## 5. Conclusions

This study examined predictors of sport coaches’ perceived performance following an 18-month mentor-based education program. The findings indicate that age, coaching hours per week, initial performance perception, and program participation significantly predicted perceived performance at post-test. However, the predictive power of the regression model was modest (R^2^ = 0.25), indicating that several other factors influence the coaches’ self-perception.

Despite the limitations, this study highlights the importance of structured mentorship, task-oriented coaching, and self-efficacy in coach development. These findings have implications for coach education programs, emphasizing the need for personalized learning pathways, feedback-driven interventions, and workload balance strategies.

## Figures and Tables

**Table 1 sports-13-00083-t001:** Pearson correlation coefficients between the investigated variables and descriptive statistics based on cross-sectional data collected from 98 coaches at pre-test.

	Variables	1	2	3	4
1	CAWA-bond	-			
2	CAWA-goal	0.25 **	-		
3	CAWA-task	0.38 *	0.67 *	-	
4	PCP	0.21 **	0.29 *	0.36 *	-
Mean		4.28	3.81	3.98	4.79
SD		0.43	0.56	0.47	0.75
Maximum		5.00	5.00	5.00	6.50
Minimum		3.00	2.00	2.75	1.50
Cronbach’s alpha		0.61	0.70	0.70	0.86

Note: * *p* < 0.01, ** *p* < 0.05.

**Table 2 sports-13-00083-t002:** Pearson correlation coefficients between the investigated variables and descriptive statistics based on cross-sectional data collected from 75 coaches during the post-test.

	Variables	1	2	3	4
1	CAWA-bond	-			
2	CAWA-goal	0.42 *	-		
3	CAWA-task	0.58 *	0.73 *	-	
4	PCP	0.28 **	0.34 *	0.40 *	-
Mean		4.24	3.81	3.97	4.80
SD		0.56	0.59	0.53	0.78
Maximum		5.00	4.75	5.00	6.50
Minimum		2.67	2.25	2.75	2.50
Cronbach’s alpha		0.69	0.70	0.79	0.86

Note: * *p* < 0.01, ** *p* < 0.05.

**Table 3 sports-13-00083-t003:** Summary of linear regression analysis for variables predicting the dependent variables (*n* = 75).

Dependent Variable	Predictors	B	t	p	R2
PCP-post	Age Coaching hours per week	0.167 0.233	1.430 2.000	0.157 0.049 **	0.04
PCP-post	Age Coaching hours per week PCP- pre	0.071 0.225 0.379	0.630 2.074 3.456	0.531 0.042 ** 0.000 *	0.17
PCP-post	Age Coaching hours per week PCP-pre CAWA-bond-development CAWA-goal-development CAWAI-task-development	0.128 0.253 0.441 0.101 −0.179 0.265	1.136 2.357 3.995 0.882 −1.443 1.959	0.260 0.021 ** 0.000 * 0.381 0.154 0.054	0.21
PCP-post	Age Coaching hours per week PCP- pre CAWA-bond-development CAWA-goal-development CAWAI-task-development Group	0.357 0.219 0.368 0.075 −0.199 0.268 −0.329	2.417 2.078 3.288 0.669 −1.652 2.047 −2.301	0.018 ** 0.042 ** 0.002 ** 0.506 0.103 0.045 ** 0.025 **	0.25

Note: * *p* < 0.001, ** *p* < 0.05.

## Data Availability

The authors report there are no competing interests to declare. Raw data were generated at the Norwegian University of Science and Technology, Department of Education and Lifelong learning. Derived data supporting the findings of this study are available from the corresponding author JAH on request.

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
