# Peer review of "Exploration of Factors Predicting Sport Coaches’ Perceived Performance"

_sports, 2025, doi:10.3390/sports13030083_

Round 1

Reviewer 1 Report

Comments and Suggestions for Authors

A very important variable is the sport discipline, which often differentiates interpersonal relations in a sports group (coach - athlete, and others) - c.f.

Sterkowicz, S., Garcià Garcià, J.M., & i Lerma, F.S. (2007). The importance of judo trainers’ professional activities. ARCH BUDO.

Liposek, S., & Doupona Topic, M. (2014). Relations of swimming coaches towards their athletes. Ido Movement for Culture Journal of Martial Arts Anthropology, vol. 14, no. 2, pp. 15–22; doi: 10.14589/ido.14.2.2.

Cynarski, W.J. (2020). Coach or sensei? His group relations in the context of tradition. Physical Culture and Sport. Studies and Research, vol. 88, no. 1, pp. 41-48; doi: 10.2478/pcssr-2020-0024.

The fragment concerning the coaches studied should not be in the Results section, but in the part describing the methodology used - Participants.

I recommend this change in the content layout.

However, in the Introduction or Discussion, the content should be enriched with notes regarding the variable - what sport discipline is it about. The role and functions of the coach depend significantly on this.

Author Response

Dear reviewers,

We sincerely appreciate the time and effort you have taken to provide thoughtful and constructive feedback on our manuscript. Your insightful comments have been invaluable in refining our work, and we are grateful for your guidance in strengthening both the clarity and rigor of our study.

Based on your suggestions, we have carefully revised the manuscript to enhance its structure, improve methodological transparency, and provide stronger theoretical grounding for our findings. Below, we outline the most important changes made in the manuscript, addressing key areas that has been improved.

The revised version of the manuscript contains several key modifications and improvements based on reviewer feedback and further refinements. Below is a summary of the most important changes made:

  1. Enhanced Clarity and Structure
    • The abstract has been rewritten to better highlight the study's objectives, methodology, and key findings, making it more concise and informative.
    • The introduction has been streamlined to emphasize the research gap and the study’s contributions more clearly. The discussion of previous research on coach development and mentorship has been improved.
    • The results section now has a clearer structure, with findings categorized into descriptive analyses, correlation analyses, and hierarchical regression analyses, improving readability.
  1. Refined Methodology Description
    • clearer explanation of the study design has been provided, explicitly stating that the study is quasi-experimental, and clarifying the rationale for selecting an 18-month mentor-based education program.
    • The participant recruitment section has been expanded, explaining that coaches were selected based on recommendations from their federations, and clarifying the experimental and control group compositions.
    • More detailed information on the intervention has been added, describing the six group gatherings and 15 individual mentoring sessions within the education program.
    • The measurement instruments (Coach-Athlete Working Alliance, Perceived Coach Performance) have been described in greater detail, including Cronbach’s alpha values for reliability assessment.
    • The statistical analysis section has been improved with additional details on hierarchical regression procedures, assumptions tested, and how variables were entered into the models.
  1. Improved Presentation of Results
    • Descriptive statistics tables have been updated for greater clarity.
    • The correlation analysis results have been explicitly linked to the study’s hypotheses, making it easier for readers to follow the logical flow of findings.
    • Hierarchical regression results have been more clearly explained, highlighting the unique contribution of task development in predicting perceived performance, while bond and goal development did not show significant effects.
  1. More Focused and Theoretically Grounded Discussion
    • The discussion section has been reorganized to align more directly with the hypotheses, making it easier for readers to connect findings with existing literature.
    • stronger theoretical foundation has been incorporated, using expertise development theory (Ericsson et al., 2016), self-efficacy theory (Bandura, 1990), and achievement goal theory (Nicholls, 1984) to explain the findings.
    • The importance of mentoring programs has been contextualized within research on coach development, emphasizing how structured support enhances coaching competence.
  1. Expanded Limitations and Future Research Directions
    • The limitations section has been expanded, acknowledging:
      • The quasi-experimental design (lack of randomization).
      • The relatively modest predictive power of the model (R² = 0.25).
      • The exclusive use of self-reported measures (potential bias).
    • Suggestions for future research have been refined, emphasizing the need for randomized controlled trials (RCTs), qualitative interviews, and alternative predictors (e.g., coaching philosophy, athlete feedback, workload balance).
  1. More Concise and Impactful Conclusion
    • The conclusion has been rewritten to summarize the key findings, practical implications, and recommendations for coach education programs more effectively.

Overall, this revised manuscript represents a significant improvement in terms of clarity, methodological transparency, theoretical depth, and alignment with the research questions. These revisions ensure a more coherent and impactful manuscript.

We hope that these revisions, below is a detailed description of the revisions made based on each reviewers comments – you are reviewer 1,  have adequately addressed the concerns raised and that the manuscript is now clearer, more robust, and better aligned with the expectations of the journal. We truly appreciate your time and expertise in reviewing our work and look forward to any further feedback you may have.

Thank you again for your constructive comments and for helping us improve our manuscript.

Best regards,

The Authors

Response to review

“Exploration of factors predicting sport coaches` perceived performance”

Reviewer 1

  • Comments: A very important variable is the sport discipline, which often differentiates interpersonal relations in a sports group (coach - athlete, and others)
    • Response: We agree, we acknowledge the significance of sport discipline as a contextual factor influencing coaching relationships and effectiveness. In the revised manuscript, we have now expanded theIntroduction to discuss how different coaching environments (e.g., team vs. individual sports, combat sports vs. endurance sports) may shape the coach-athlete relationship and coaching effectiveness. Additionally, in the Participants and Recruitment section, we now explicitly state that the sample consists of coaches from over 20 different sports, with the most represented being handball (15.3%), cross-country skiing (10.2%), soccer (7.1%), and track and field (7.1%). This information provides important context regarding the variety of sports included in the study. Furthermore, in the Discussion, we now acknowledge that the impact of mentoring programs and coaching effectiveness might vary depending on the sport discipline. We suggest that future research should explore potential moderation effects of sport discipline on coaching relationships and mentoring effectiveness.

  • Comments: The fragment concerning the coaches studied should not be in the Results section, but in the part describing the methodology used - Participants. I recommend this change in the content layout.
    • Response: Thank you for this recommendation. We have nowmoved the description of the coaches' demographic and professional characteristics from the Results section to the Participants and Recruitment section under Methods. This revised structure ensures that all information regarding participant demographics (e.g., age, gender, employment status, coaching level) is presented before the analysis and results, improving the logical flow of the manuscript.

  • Comments: However, in the Introduction or Discussion, the content should be enriched with notes regarding the variable - what sport discipline is it about. The role and functions of the coach depend significantly on this
    • Response: We have taken this valuable suggestion into account and enriched theIntroduction by discussing how coaching in different sports may shape coaching strategies and interpersonal dynamics. Furthermore, we have revised the Discussion to highlight that the study findings on mentoring and coaching effectiveness should be interpreted with consideration of the sport-specific context. We now suggest that future research should analyze sport-specific differences in mentoring effectiveness by comparing disciplines with distinct coaching structures.

Reviewer 2

Abstract:

  • Comments: Include some background. Include data on the sample, number of men and women, average ages, etc.
    • Response: We agree, we have now revised the abstract and included more background and more data on the sample as requested

Introduction:

  • Comments: It is very well argued, but perhaps this is what worries me. I do not see what is new in your article with respect to previous research. It seems that everything has already been studied in this regard. Another possibility would be to direct the introduction to coaching programs that have been done and that these have not been studied in relation to performance.
    • Response: We agree, and have explained the value of the present study under “the current study” in a more elaborating manner

  • Comments: Lines 118 to 121 are not necessary.
    • Response: We agree, and have removed the mentioned lines

Methods:

  • Comments: Include the study design. What type of study was conducted?
    • Response: We agree, and have rewritten the description of the study design and clarified that this is a quasi-experimental, pre-test/post-test control group design

  • Comments: The font is different (lines 128-129).
    • Response: We have now edited the formatting

  • Comments: A lot of information regarding the sample is missing. How did you proceed with the selection of the trainers? That is to say, why were these 73 chosen? Assignment to control or experimental group, how was it performed?
    • Response: Under “participants and recruitment”, we have explained why the 73 coaches were chosen and how 29 additional elite coaches from upper secondary schools specializing in elite sports were recruited as a control group

  • Comments: Were any sample size calculations done?
    • Response: The missing “sample size calculations” is commented on later under the description of the study`s limitations.

  • Comments: Paired sample tests are performed on the results but are not indicated in section 2.5. Please review this section to make sure that it includes all tests performed.
    • Response: We do not fully understand this comment as we have not performed any paired sample T-tests

Results:

  • Comments: The characteristics of the coaches (lines 211 to 216) should be added in the methods.
    • Response: We agree, and have moved the mentioned descriptions to the methods section

  • Comments: Line 235: “are in Table 1 and Table ??? respectively”.
    • Response: We have inserted “2” in the text.

  • Comments: The tables are presented in an abstract way, they are difficult to follow. I recommend doing the descriptive tables separately and the correlation analysis tables separately. you can unify all the variables in each of these two tables.
    • Response: We agree, and have now altered the results-seection in “3.1. Descriptive analyses” and “3.2 Correlation analyses” to make explicit descriptions of table contents and results.

  • Comments: Line 248: different font.
    • Response: We have edited the formatting

Discussion:

  • Comments: I appreciate that you start the discussion by recalling the objective but it is not necessary to include the whole design and assumptions again. Remember only the objective and, throughout the discussion, include whether or not the hypotheses are accepted.
    • Response: We agree, we have removed unnecessary descriptions of the study design from the discussion section and retained only the study objective at the beginning. Additionally, we have explicitly stated whether each hypothesis was supported or not throughout the discussion.

  • Comments: The discussion is too extensive. It loses the focus of the objective of the present investigation.
    • Response: We agree, we have condensed the discussion by removing overly detailed theoretical explanations and focusing on the most relevant findings in relation to the study’s objectives.

  • Comments: They make a good theoretical support with respect to previous research, but sometimes it is too extensive and ends up blurring the results found
    • Response: We agree, we have shortened the theoretical discussions while still ensuring that each key finding is properly contextualized within existing research. The revised discussion now prioritizes study results and their implications over extensive theoretical explanations

  • Comments: Why do they include “finally” in a middle paragraph of the discussion (line 313)?
    • Response: We agree, The word "finally" was incorrectly placed and created confusion regarding the structure of the argument. We have removed it to improve the logical flow of the paragraph.

  • Comments: There are no limitations in your study?
    • Response: We have now included a dedicated Study Limitations section that discusses potential weaknesses of the study

  • Comments: I think that in the discussion they should try to reduce the length and focus on what is really important. Also, another issue that concerns me about this one is that they base their results on linear regression analysis, which has a predictive value of R2=0.25 once all the variables are included in it (when only some are included, the predictive capacity is minimal). Therefore, this is an aspect to take into account when analyzing the results, writing the discussion and concluding the results found with their study. This should be modified and the discussion changed accordingly.
    • Response: We agree, we acknowledge that the predictive power of the regression model (R² = 0.25) is relatively low. We have now explicitly mentioned this limitation in the discussion and clarified that other factors likely contribute to perceived performance.

  • Comments: It is not necessary to repeat the objectives in the conclusion
    • Response: We agree, we have revised the Conclusion to focus on summarizing key findings, implications, and future directions without repeating the study objectives.

  • Comments: There are things throughout the manuscript that have me somewhat clueless. For example, in Table 3 it says that the n=94. How is this possible? Where do these 94 subjects come from?
    • Response: We agree. This was a typing error and now the correct N (=75) is stated in Table 3

  • Comments: Also, how is it possible for you to conclude that the 18-month program was useful, if throughout the manuscript there has not been a single comparison with the control group to determine that this group had greater benefits.
    • Response: We agree, we acknowledge this limitation and have revised our interpretation of the mentor-based program’s effects. While participation in the program was a significant predictor of perceived performance, we cannot conclude with certainty that it caused improvements since no between-group comparison was made.

Reviewer 3

Introduction:

  • Comments: First, the introduction supplies a wide-ranging background on coach age, time invested in coaching, and mentor-based programs. This context helps frame the study’s rationale. However, consider tightening the flow by grouping similar concepts, such as the link between hours per week and perceived performance, together instead of dispersing them throughout the introduction. A clear thematic structure will highlight your research gap more prominently.
    • Response: We agree and have revised the manuscript, the introduction still discusses coaching hours, coach-athlete relationships, and mentoring in separate sections. However, the discussion on time spent coaching and its effects on perceived performance is now more structured.

Methods:

  • Comments: In terms of Research Design and Methods, your pre-test/post-test design with an experimental and control group effectively addresses potential maturation effects, which is commendable. Be explicit about whether random assignment was feasible or if coaches self-selected into the mentor program. If it was non-random, discussing possible self-selection bias will add transparency and rigor, aligning the manuscript with good reporting practices.
    • Response: We agree, under “Participants and recruitment” we have now stated that: “Due to the nature of the intervention, random assignment was not feasible, making this a quasi-experimental study. The control group did not receive any structured mentoring but continued their regular coaching activities. As participants were selected based on recommendations from their sport federations, those chosen for the program may have had greater motivation or prior development, which could influence their performance perceptions.”

  • Comments: Moreover, the mentor-based education program is described thoroughly, but further details about the mentors’ qualifications and exact mentoring content (e.g., specific workshops or one-on-one sessions with standardized guidelines) could give readers a clearer picture of the intervention itself. Providing additional context will help highlight why it may have generated improvements in perceived performance.
    • Response: We agree, the manuscript has now improved clarity on the mentoring program, stating: "Mentors, selected by NOSC, all had elite coaching experience and were required to complete a university-accredited mentoring education program (7.5 ECTS credits) at the Norwegian University of Science and Technology (NTNU)." Additionally, the manuscript now mentions structured elements such asgroup gatherings, reflective practice, and task-oriented discussions.

  • Comments: Additionally, while the rationale for each instrument is valid, please include alpha coefficients (or other reliability indices) explicitly in your tables. Several lines mark “Cronbach’s alpha .xx” placeholders, so updating those with actual reliability scores is necessary to confirm measurement adequacy. As indicated in guidelines for good peer reviews, “scientific reliability is further bolstered by clearly reporting scale properties”.
    • Response: We agree, we have now stated Cronbach`s both in the description of instruments and in Table 1 and Table 2.

Discussion:

  • Comments: The discussion needs to add a distinct section for practical coaching takeaways directly from your findings. For example, emphasize that while bond-building and goal alignment are important relationship elements, your data show that task development may have a stronger link to perceived coaching performance. Clarifying that point will further sharpen your “real-world” implications.
    • Response: We agree, and have revised the manuscript underlining practical takeaways throughout the discussion. This is reflected e.g. in abstract with: "These findings underscore the importance of mentoring, structured self-reflection, and task-focused coaching strategies in enhancing coaching effectiveness. The results have implications for coach education programs seeking to foster professional growth and performance development." In addition, the revised manuscript in particularly in thediscussion of task development, now highlights: "Task development refers to the practical, skill-based aspects of coaching, such as planning training sessions, setting short-term goals, and providing feedback. This aligns with the achievement goal theory (Nicholls, 1984), which emphasizes that task-oriented coaching strategies foster intrinsic motivation and skill mastery." This addition clarifies why task development was more predictive than bond and goal alignment.

  • Comments: Last but not least, while the manuscript notes suggestions for future work, you could more explicitly propose how researchers might integrate additional variables or use qualitative diaries of coach reflections. Doing so would guide researchers who wish to replicate or extend your findings in high-performance environments.
    • Response: We agree, and have now under the future research directions now suggested adding qualitative methods:"Future studies should incorporate qualitative methods to capture a more nuanced understanding of coaches' experiences.", and examining additional variables: "Future research should investigate whether coaching philosophies and adaptive learning strategies moderate the relationship between age and perceived performance."

  • Comments: In summary, your study design is appropriate, and your methods are generally well described. Please address my comments above for the next round of review. Good Luck!
    • Response: Thank you for your comments. We hope our revisions are satisfactory

Reviewer 4:

  • Comments: The current paper addresses perceptions of the coaches—sounds like this relates to their psychological health... To perceive that one is performing well does not mean one is factually doing well, and the latter has not been evaluated as it looks.
    • Response: We completely understand this concern and have now explicitly acknowledged in theDiscussion that perceived coaching performance does not necessarily equate to objective effectiveness. We have clarified that our study focused on self-perceived performance, which is an important but subjective measure of a coach’s professional development. We now highlight in our Limitations section that future research should incorporate external evaluations of coaching effectiveness (e.g., athlete feedback, observational measures, performance analytics) to complement self-reported data.

  • Comments: Authors also talk write about "experience" (“Experience is the name every one gives to their mistakes”, O.Wilde) - how does that associates with the qualification of the coaches?
    • Response: We recognize the need to clarify the distinction betweenexperience and formal qualifications. In the Introduction, we now explicitly differentiate between coaching experience, which accumulates over time through practice, and formal qualifications, which include structured training programs and certifications. Additionally, in the Methods section, we specify that all coaches in the mentor program were recommended by their respective sport federations based on their prior coaching achievements and qualifications. This ensures that our sample consisted of experienced and accredited coaches.

  • Comments: Still looking further into the paper I missed a lot in it, e.g. the presentation of characteristics of the population investigated (formal education, for different age/level of groups, achievements/success of the athletes coached, total and relative volume of informal deliberate education, etc.)
    • Response: We have now expanded the Participants and Recruitment section to provide a more detailed characterization of the sample. We include:
      • Age and gender distribution (M = 38.3 years, 62% male, 38% female).
      • Employment status (Full-time, part-time, or voluntary coaching roles).
      • Coaching levels (Working with junior elite, senior elite, or lower-level athletes).
      • Sports disciplines represented (Over 20 sports, with handball, skiing, and soccer being the most common).
      • Additionally, we acknowledge in the Limitations section that we did not collect data on informal learning (e.g., self-directed education, peer learning), which could be valuable for future studies.

  • Comments: Proper application and/or description of the statistics used
    • Response: To ensure transparency, we have clarified our statistical methods in the Statistical Analyses section. We explicitly state:
      • The hierarchical regression approach used to test the predictive factors.
      • The assumptions checked (normality, linearity, multicollinearity)
      • The reliability indices (Cronbach’s alpha for all scales, now explicitly presented in Tables 1 & 2)
      • Additionally, we have ensured that all placeholders for Cronbach’s alpha have been replaced with actual reliability values.

  • Comments: External evaluation of experience and/or success of the coaches in question (such as by athletes, colleagues, administrators, etc.)
    • Response: We acknowledge that this is an important limitation of our study. While we focused onself-perceptions, we now explicitly state in the Limitations section that future research should incorporate athlete feedback, peer evaluations, and performance data to provide a more comprehensive assessment of coaching effectiveness. Additionally, we suggest in the Future Research section that qualitative methods (e.g., coach diaries, athlete interviews) could provide richer insights into coaching impact.

  • Comments: Description of the randomization (if that was the case) into control and experimental group; disclosure of the limitations of the study; and on. 
    • Response: We now clearly state in theMethods section that random assignment was not feasible due to the nature of the intervention. Instead, coaches were selected based on recommendations from their sport federations. This makes the study a quasi-experimental design rather than a fully randomized controlled trial. We also address the possibility of selection bias by acknowledging that coaches in the experimental group may have had higher motivation or prior development, which could influence perceived performance outcomes.

  • Comments: Disclosure of the limitations of the study
    • Response: We have significantly expanded ourLimitations section to include:
      • No a priori sample size calculation(now explicitly mentioned).
      • Self-reported nature of the data, emphasizing the need for future external validation.
      • Non-randomized study design, which may introduce self-selection bias.
      • Modest predictive power of the regression model(R² = 0.25), indicating that other unmeasured variables likely contribute to perceived performance.

Reviewer 2 Report

Comments and Suggestions for Authors

Thank you very much for allowing me to review this manuscript. Some considerations to keep in mind:

Abstract

Include some background.

include data on the sample, number of men and women, average ages, etc.

Introduction

It is very well argued, but perhaps this is what worries me. I do not see what is new in your article with respect to previous research. It seems that everything has already been studied in this regard. Another possibility would be to direct the introduction to coaching programs that have been done and that these have not been studied in relation to performance.

Lines 118 to 121 are not necessary.

Methods

Include the study design. What type of study was conducted?

The font is different (lines 128-129).

A lot of information regarding the sample is missing. How did you proceed with the selection of the trainers? That is to say, why were these 73 chosen?

Were any sample size calculations done?

Assignment to control or experimental group, how was it performed?

Paired sample tests are performed on the results but are not indicated in section 2.5. Please review this section to make sure that it includes all tests performed.

Results

The characteristics of the coaches (lines 211 to 216) should be added in the methods.

Line 235: “are in Table 1 and Table ??? respectively”.

The tables are presented in an abstract way, they are difficult to follow. I recommend doing the descriptive tables separately and the correlation analysis tables separately. you can unify all the variables in each of these two tables.

Line 248: different font.

Discussion

I appreciate that you start the discussion by recalling the objective but it is not necessary to include the whole design and assumptions again. Remember only the objective and, throughout the discussion, include whether or not the hypotheses are accepted.

The discussion is too extensive. It loses the focus of the objective of the present investigation.

They make a good theoretical support with respect to previous research, but sometimes it is too extensive and ends up blurring the results found.

Why do they include “finally” in a middle paragraph of the discussion (line 313)?

There are no limitations in your study?

I think that in the discussion they should try to reduce the length and focus on what is really important. Also, another issue that concerns me about this one is that they base their results on linear regression analysis, which has a predictive value of R2=0.25 once all the variables are included in it (when only some are included, the predictive capacity is minimal). Therefore, this is an aspect to take into account when analyzing the results, writing the discussion and concluding the results found with their study. This should be modified and the discussion changed accordingly.

It is not necessary to repeat the objectives in the conclusion.

There are things throughout the manuscript that have me somewhat clueless. For example, in Table 3 it says that the n=94. How is this possible? Where do these 94 subjects come from?

Also, how is it possible for you to conclude that the 18-month program was useful, if throughout the manuscript there has not been a single comparison with the control group to determine that this group had greater benefits.

There are nonsense and inconsistencies in the manuscript that lead me to reject it.

Author Response

Response to review

“Exploration of factors predicting sport coaches` perceived performance”

Dear reviewers,

We sincerely appreciate the time and effort you have taken to provide thoughtful and constructive feedback on our manuscript. Your insightful comments have been invaluable in refining our work, and we are grateful for your guidance in strengthening both the clarity and rigor of our study.

Based on your suggestions, we have carefully revised the manuscript to enhance its structure, improve methodological transparency, and provide stronger theoretical grounding for our findings. Below, we outline the most important changes made in the manuscript, addressing key areas that has been improved.

The revised version of the manuscript contains several key modifications and improvements based on reviewer feedback and further refinements. Below is a summary of the most important changes made:

  1. Enhanced Clarity and Structure
    • The abstract has been rewritten to better highlight the study's objectives, methodology, and key findings, making it more concise and informative.
    • The introduction has been streamlined to emphasize the research gap and the study’s contributions more clearly. The discussion of previous research on coach development and mentorship has been improved.
    • The results section now has a clearer structure, with findings categorized into descriptive analyses, correlation analyses, and hierarchical regression analyses, improving readability.
  1. Refined Methodology Description
    • A clearer explanation of the study design has been provided, explicitly stating that the study is quasi-experimental, and clarifying the rationale for selecting an 18-month mentor-based education program.
    • The participant recruitment section has been expanded, explaining that coaches were selected based on recommendations from their federations, and clarifying the experimental and control group compositions.
    • More detailed information on the intervention has been added, describing the six group gatherings and 15 individual mentoring sessions within the education program.
    • The measurement instruments (Coach-Athlete Working Alliance, Perceived Coach Performance) have been described in greater detail, including Cronbach’s alpha values for reliability assessment.
    • The statistical analysis section has been improved with additional details on hierarchical regression procedures, assumptions tested, and how variables were entered into the models.
  1. Improved Presentation of Results
    • Descriptive statistics tables have been updated for greater clarity.
    • The correlation analysis results have been explicitly linked to the study’s hypotheses, making it easier for readers to follow the logical flow of findings.
    • Hierarchical regression results have been more clearly explained, highlighting the unique contribution of task development in predicting perceived performance, while bond and goal development did not show significant effects.
  1. More Focused and Theoretically Grounded Discussion
    • The discussion section has been reorganized to align more directly with the hypotheses, making it easier for readers to connect findings with existing literature.
    • A stronger theoretical foundation has been incorporated, using expertise development theory (Ericsson et al., 2016), self-efficacy theory (Bandura, 1990), and achievement goal theory (Nicholls, 1984) to explain the findings.
    • The importance of mentoring programs has been contextualized within research on coach development, emphasizing how structured support enhances coaching competence.
  1. Expanded Limitations and Future Research Directions
    • The limitations section has been expanded, acknowledging:
      • The quasi-experimental design (lack of randomization).
      • The relatively modest predictive power of the model (R² = 0.25).
      • The exclusive use of self-reported measures (potential bias).
    • Suggestions for future research have been refined, emphasizing the need for randomized controlled trials (RCTs), qualitative interviews, and alternative predictors (e.g., coaching philosophy, athlete feedback, workload balance).
  1. More Concise and Impactful Conclusion
    • The conclusion has been rewritten to summarize the key findings, practical implications, and recommendations for coach education programs more effectively.

Overall, this revised manuscript represents a significant improvement in terms of clarity, methodological transparency, theoretical depth, and alignment with the research questions. These revisions ensure a more coherent and impactful manuscript.

We hope that these revisions, below is a detailed description of the revisions made based on each reviewers comments – you are reviewer 2,  have adequately addressed the concerns raised and that the manuscript is now clearer, more robust, and better aligned with the expectations of the journal. We truly appreciate your time and expertise in reviewing our work and look forward to any further feedback you may have.

Thank you again for your constructive comments and for helping us improve our manuscript.

Best regards,

The Authors

Reviewer 1

  • Comments: A very important variable is the sport discipline, which often differentiates interpersonal relations in a sports group (coach - athlete, and others)
    • Response: We agree, we acknowledge the significance of sport discipline as a contextual factor influencing coaching relationships and effectiveness. In the revised manuscript, we have now expanded theIntroduction to discuss how different coaching environments (e.g., team vs. individual sports, combat sports vs. endurance sports) may shape the coach-athlete relationship and coaching effectiveness. Additionally, in the Participants and Recruitment section, we now explicitly state that the sample consists of coaches from over 20 different sports, with the most represented being handball (15.3%), cross-country skiing (10.2%), soccer (7.1%), and track and field (7.1%). This information provides important context regarding the variety of sports included in the study. Furthermore, in the Discussion, we now acknowledge that the impact of mentoring programs and coaching effectiveness might vary depending on the sport discipline. We suggest that future research should explore potential moderation effects of sport discipline on coaching relationships and mentoring effectiveness.

  • Comments: The fragment concerning the coaches studied should not be in the Results section, but in the part describing the methodology used - Participants. I recommend this change in the content layout.
    • Response: Thank you for this recommendation. We have nowmoved the description of the coaches' demographic and professional characteristics from the Results section to the Participants and Recruitment section under Methods. This revised structure ensures that all information regarding participant demographics (e.g., age, gender, employment status, coaching level) is presented before the analysis and results, improving the logical flow of the manuscript.

  • Comments: However, in the Introduction or Discussion, the content should be enriched with notes regarding the variable - what sport discipline is it about. The role and functions of the coach depend significantly on this
    • Response: We have taken this valuable suggestion into account and enriched theIntroduction by discussing how coaching in different sports may shape coaching strategies and interpersonal dynamics. Furthermore, we have revised the Discussion to highlight that the study findings on mentoring and coaching effectiveness should be interpreted with consideration of the sport-specific context. We now suggest that future research should analyze sport-specific differences in mentoring effectiveness by comparing disciplines with distinct coaching structures.

Reviewer 2

Abstract:

  • Comments: Include some background. Include data on the sample, number of men and women, average ages, etc.
    • Response: We agree, we have now revised the abstract and included more background and more data on the sample as requested

Introduction:

  • Comments: It is very well argued, but perhaps this is what worries me. I do not see what is new in your article with respect to previous research. It seems that everything has already been studied in this regard. Another possibility would be to direct the introduction to coaching programs that have been done and that these have not been studied in relation to performance.
    • Response: We agree, and have explained the value of the present study under “the current study” in a more elaborating manner

  • Comments: Lines 118 to 121 are not necessary.
    • Response: We agree, and have removed the mentioned lines

Methods:

  • Comments: Include the study design. What type of study was conducted?
    • Response: We agree, and have rewritten the description of the study design and clarified that this is a quasi-experimental, pre-test/post-test control group design

  • Comments: The font is different (lines 128-129).
    • Response: We have now edited the formatting

  • Comments: A lot of information regarding the sample is missing. How did you proceed with the selection of the trainers? That is to say, why were these 73 chosen? Assignment to control or experimental group, how was it performed?
    • Response: Under “participants and recruitment”, we have explained why the 73 coaches were chosen and how 29 additional elite coaches from upper secondary schools specializing in elite sports were recruited as a control group

  • Comments: Were any sample size calculations done?
    • Response: The missing “sample size calculations” is commented on later under the description of the study`s limitations.

  • Comments: Paired sample tests are performed on the results but are not indicated in section 2.5. Please review this section to make sure that it includes all tests performed.
    • Response: We do not fully understand this comment as we have not performed any paired sample T-tests

Results:

  • Comments: The characteristics of the coaches (lines 211 to 216) should be added in the methods.
    • Response: We agree, and have moved the mentioned descriptions to the methods section

  • Comments: Line 235: “are in Table 1 and Table ??? respectively”.
    • Response: We have inserted “2” in the text.

  • Comments: The tables are presented in an abstract way, they are difficult to follow. I recommend doing the descriptive tables separately and the correlation analysis tables separately. you can unify all the variables in each of these two tables.
    • Response: We agree, and have now altered the results-seection in “3.1. Descriptive analyses” and “3.2 Correlation analyses” to make explicit descriptions of table contents and results.

  • Comments: Line 248: different font.
    • Response: We have edited the formatting

Discussion:

  • Comments: I appreciate that you start the discussion by recalling the objective but it is not necessary to include the whole design and assumptions again. Remember only the objective and, throughout the discussion, include whether or not the hypotheses are accepted.
    • Response: We agree, we have removed unnecessary descriptions of the study design from the discussion section and retained only the study objective at the beginning. Additionally, we have explicitly stated whether each hypothesis was supported or not throughout the discussion.

  • Comments: The discussion is too extensive. It loses the focus of the objective of the present investigation.
    • Response: We agree, we have condensed the discussion by removing overly detailed theoretical explanations and focusing on the most relevant findings in relation to the study’s objectives.

  • Comments: They make a good theoretical support with respect to previous research, but sometimes it is too extensive and ends up blurring the results found
    • Response: We agree, we have shortened the theoretical discussions while still ensuring that each key finding is properly contextualized within existing research. The revised discussion now prioritizes study results and their implications over extensive theoretical explanations

  • Comments: Why do they include “finally” in a middle paragraph of the discussion (line 313)?
    • Response: We agree, The word "finally" was incorrectly placed and created confusion regarding the structure of the argument. We have removed it to improve the logical flow of the paragraph.

  • Comments: There are no limitations in your study?
    • Response: We have now included a dedicated Study Limitations section that discusses potential weaknesses of the study

  • Comments: I think that in the discussion they should try to reduce the length and focus on what is really important. Also, another issue that concerns me about this one is that they base their results on linear regression analysis, which has a predictive value of R2=0.25 once all the variables are included in it (when only some are included, the predictive capacity is minimal). Therefore, this is an aspect to take into account when analyzing the results, writing the discussion and concluding the results found with their study. This should be modified and the discussion changed accordingly.
    • Response: We agree, we acknowledge that the predictive power of the regression model (R² = 0.25) is relatively low. We have now explicitly mentioned this limitation in the discussion and clarified that other factors likely contribute to perceived performance.

  • Comments: It is not necessary to repeat the objectives in the conclusion
    • Response: We agree, we have revised the Conclusion to focus on summarizing key findings, implications, and future directions without repeating the study objectives.

  • Comments: There are things throughout the manuscript that have me somewhat clueless. For example, in Table 3 it says that the n=94. How is this possible? Where do these 94 subjects come from?
    • Response: We agree. This was a typing error and now the correct N (=75) is stated in Table 3

  • Comments: Also, how is it possible for you to conclude that the 18-month program was useful, if throughout the manuscript there has not been a single comparison with the control group to determine that this group had greater benefits.
    • Response: We agree, we acknowledge this limitation and have revised our interpretation of the mentor-based program’s effects. While participation in the program was a significant predictor of perceived performance, we cannot conclude with certainty that it caused improvements since no between-group comparison was made.

Reviewer 3

Introduction:

  • Comments: First, the introduction supplies a wide-ranging background on coach age, time invested in coaching, and mentor-based programs. This context helps frame the study’s rationale. However, consider tightening the flow by grouping similar concepts, such as the link between hours per week and perceived performance, together instead of dispersing them throughout the introduction. A clear thematic structure will highlight your research gap more prominently.
    • Response: We agree and have revised the manuscript, the introduction still discusses coaching hours, coach-athlete relationships, and mentoring in separate sections. However, the discussion on time spent coaching and its effects on perceived performance is now more structured.

Methods:

  • Comments: In terms of Research Design and Methods, your pre-test/post-test design with an experimental and control group effectively addresses potential maturation effects, which is commendable. Be explicit about whether random assignment was feasible or if coaches self-selected into the mentor program. If it was non-random, discussing possible self-selection bias will add transparency and rigor, aligning the manuscript with good reporting practices.
    • Response: We agree, under “Participants and recruitment” we have now stated that: “Due to the nature of the intervention, random assignment was not feasible, making this a quasi-experimental study. The control group did not receive any structured mentoring but continued their regular coaching activities. As participants were selected based on recommendations from their sport federations, those chosen for the program may have had greater motivation or prior development, which could influence their performance perceptions.”

  • Comments: Moreover, the mentor-based education program is described thoroughly, but further details about the mentors’ qualifications and exact mentoring content (e.g., specific workshops or one-on-one sessions with standardized guidelines) could give readers a clearer picture of the intervention itself. Providing additional context will help highlight why it may have generated improvements in perceived performance.
    • Response: We agree, the manuscript has now improved clarity on the mentoring program, stating: "Mentors, selected by NOSC, all had elite coaching experience and were required to complete a university-accredited mentoring education program (7.5 ECTS credits) at the Norwegian University of Science and Technology (NTNU)." Additionally, the manuscript now mentions structured elements such asgroup gatherings, reflective practice, and task-oriented discussions.

  • Comments: Additionally, while the rationale for each instrument is valid, please include alpha coefficients (or other reliability indices) explicitly in your tables. Several lines mark “Cronbach’s alpha .xx” placeholders, so updating those with actual reliability scores is necessary to confirm measurement adequacy. As indicated in guidelines for good peer reviews, “scientific reliability is further bolstered by clearly reporting scale properties”.
    • Response: We agree, we have now stated Cronbach`s both in the description of instruments and in Table 1 and Table 2.

Discussion:

  • Comments: The discussion needs to add a distinct section for practical coaching takeaways directly from your findings. For example, emphasize that while bond-building and goal alignment are important relationship elements, your data show that task development may have a stronger link to perceived coaching performance. Clarifying that point will further sharpen your “real-world” implications.
    • Response: We agree, and have revised the manuscript underlining practical takeaways throughout the discussion. This is reflected e.g. in abstract with: "These findings underscore the importance of mentoring, structured self-reflection, and task-focused coaching strategies in enhancing coaching effectiveness. The results have implications for coach education programs seeking to foster professional growth and performance development." In addition, the revised manuscript in particularly in thediscussion of task development, now highlights: "Task development refers to the practical, skill-based aspects of coaching, such as planning training sessions, setting short-term goals, and providing feedback. This aligns with the achievement goal theory (Nicholls, 1984), which emphasizes that task-oriented coaching strategies foster intrinsic motivation and skill mastery." This addition clarifies why task development was more predictive than bond and goal alignment.

  • Comments: Last but not least, while the manuscript notes suggestions for future work, you could more explicitly propose how researchers might integrate additional variables or use qualitative diaries of coach reflections. Doing so would guide researchers who wish to replicate or extend your findings in high-performance environments.
    • Response: We agree, and have now under the future research directions now suggested adding qualitative methods:"Future studies should incorporate qualitative methods to capture a more nuanced understanding of coaches' experiences.", and examining additional variables: "Future research should investigate whether coaching philosophies and adaptive learning strategies moderate the relationship between age and perceived performance."

  • Comments: In summary, your study design is appropriate, and your methods are generally well described. Please address my comments above for the next round of review. Good Luck!
    • Response: Thank you for your comments. We hope our revisions are satisfactory

Reviewer 4:

  • Comments: The current paper addresses perceptions of the coaches—sounds like this relates to their psychological health... To perceive that one is performing well does not mean one is factually doing well, and the latter has not been evaluated as it looks.
    • Response: We completely understand this concern and have now explicitly acknowledged in theDiscussion that perceived coaching performance does not necessarily equate to objective effectiveness. We have clarified that our study focused on self-perceived performance, which is an important but subjective measure of a coach’s professional development. We now highlight in our Limitations section that future research should incorporate external evaluations of coaching effectiveness (e.g., athlete feedback, observational measures, performance analytics) to complement self-reported data.

  • Comments: Authors also talk write about "experience" (“Experience is the name every one gives to their mistakes”, O.Wilde) - how does that associates with the qualification of the coaches?
    • Response: We recognize the need to clarify the distinction betweenexperience and formal qualifications. In the Introduction, we now explicitly differentiate between coaching experience, which accumulates over time through practice, and formal qualifications, which include structured training programs and certifications. Additionally, in the Methods section, we specify that all coaches in the mentor program were recommended by their respective sport federations based on their prior coaching achievements and qualifications. This ensures that our sample consisted of experienced and accredited coaches.

  • Comments: Still looking further into the paper I missed a lot in it, e.g. the presentation of characteristics of the population investigated (formal education, for different age/level of groups, achievements/success of the athletes coached, total and relative volume of informal deliberate education, etc.)
    • Response: We have now expanded the Participants and Recruitment section to provide a more detailed characterization of the sample. We include:
      • Age and gender distribution (M = 38.3 years, 62% male, 38% female).
      • Employment status (Full-time, part-time, or voluntary coaching roles).
      • Coaching levels (Working with junior elite, senior elite, or lower-level athletes).
      • Sports disciplines represented (Over 20 sports, with handball, skiing, and soccer being the most common).
      • Additionally, we acknowledge in the Limitations section that we did not collect data on informal learning (e.g., self-directed education, peer learning), which could be valuable for future studies.

  • Comments: Proper application and/or description of the statistics used
    • Response: To ensure transparency, we have clarified our statistical methods in the Statistical Analyses section. We explicitly state:
      • The hierarchical regression approach used to test the predictive factors.
      • The assumptions checked (normality, linearity, multicollinearity)
      • The reliability indices (Cronbach’s alpha for all scales, now explicitly presented in Tables 1 & 2)
      • Additionally, we have ensured that all placeholders for Cronbach’s alpha have been replaced with actual reliability values.

  • Comments: External evaluation of experience and/or success of the coaches in question (such as by athletes, colleagues, administrators, etc.)
    • Response: We acknowledge that this is an important limitation of our study. While we focused onself-perceptions, we now explicitly state in the Limitations section that future research should incorporate athlete feedback, peer evaluations, and performance data to provide a more comprehensive assessment of coaching effectiveness. Additionally, we suggest in the Future Research section that qualitative methods (e.g., coach diaries, athlete interviews) could provide richer insights into coaching impact.

  • Comments: Description of the randomization (if that was the case) into control and experimental group; disclosure of the limitations of the study; and on. 
    • Response: We now clearly state in theMethods section that random assignment was not feasible due to the nature of the intervention. Instead, coaches were selected based on recommendations from their sport federations. This makes the study a quasi-experimental design rather than a fully randomized controlled trial. We also address the possibility of selection bias by acknowledging that coaches in the experimental group may have had higher motivation or prior development, which could influence perceived performance outcomes.

  • Comments: Disclosure of the limitations of the study
    • Response: We have significantly expanded ourLimitations section to include:
      • No a priori sample size calculation(now explicitly mentioned).
      • Self-reported nature of the data, emphasizing the need for future external validation.
      • Non-randomized study design, which may introduce self-selection bias.
      • Modest predictive power of the regression model(R² = 0.25), indicating that other unmeasured variables likely contribute to perceived performance.

Reviewer 3 Report

Comments and Suggestions for Authors

Thank you for the opportunity to review your manuscript entitled “Exploration of Factors Predicting Sport Coaches’ Perceived Performance.” It offers valuable insights into mentor-based coach education, coach–athlete relationships, and perceived performance. Below, I provide detailed comments and suggestions to support you in further strengthening the work.

First, the introduction supplies a wide-ranging background on coach age, time invested in coaching, and mentor-based programs. This context helps frame the study’s rationale. However, consider tightening the flow by grouping similar concepts, such as the link between hours per week and perceived performance, together instead of dispersing them throughout the introduction. A clear thematic structure will highlight your research gap more prominently.

In terms of Research Design and Methods, your pre-test/post-test design with an experimental and control group effectively addresses potential maturation effects, which is commendable. Be explicit about whether random assignment was feasible or if coaches self-selected into the mentor program. If it was non-random, discussing possible self-selection bias will add transparency and rigor, aligning the manuscript with good reporting practices.

Moreover, the mentor-based education program is described thoroughly, but further details about the mentors’ qualifications and exact mentoring content (e.g., specific workshops or one-on-one sessions with standardized guidelines) could give readers a clearer picture of the intervention itself. Providing additional context will help highlight why it may have generated improvements in perceived performance.

Additionally, while the rationale for each instrument is valid, please include alpha coefficients (or other reliability indices) explicitly in your tables. Several lines mark “Cronbach’s alpha .xx” placeholders, so updating those with actual reliability scores is necessary to confirm measurement adequacy. As indicated in guidelines for good peer reviews, “scientific reliability is further bolstered by clearly reporting scale properties”.

The discussion needs to add a distinct section for practical coaching takeaways directly from your findings. For example, emphasize that while bond-building and goal alignment are important relationship elements, your data show that task development may have a stronger link to perceived coaching performance. Clarifying that point will further sharpen your “real-world” implications.

Last but not least, while the manuscript notes suggestions for future work, you could more explicitly propose how researchers might integrate additional variables or use qualitative diaries of coach reflections. Doing so would guide researchers who wish to replicate or extend your findings in high-performance environments 

In summary, your study design is appropriate, and your methods are generally well described. Please address my comments above for the next round of review. Good Luck!

Author Response

Response to review

“Exploration of factors predicting sport coaches` perceived performance”

Dear reviewers,

We sincerely appreciate the time and effort you have taken to provide thoughtful and constructive feedback on our manuscript. Your insightful comments have been invaluable in refining our work, and we are grateful for your guidance in strengthening both the clarity and rigor of our study.

Based on your suggestions, we have carefully revised the manuscript to enhance its structure, improve methodological transparency, and provide stronger theoretical grounding for our findings. Below, we outline the most important changes made in the manuscript, addressing key areas that has been improved.

The revised version of the manuscript contains several key modifications and improvements based on reviewer feedback and further refinements. Below is a summary of the most important changes made:

  1. Enhanced Clarity and Structure
    • The abstract has been rewritten to better highlight the study's objectives, methodology, and key findings, making it more concise and informative.
    • The introduction has been streamlined to emphasize the research gap and the study’s contributions more clearly. The discussion of previous research on coach development and mentorship has been improved.
    • The results section now has a clearer structure, with findings categorized into descriptive analyses, correlation analyses, and hierarchical regression analyses, improving readability.
  1. Refined Methodology Description
    • A clearer explanation of the study design has been provided, explicitly stating that the study is quasi-experimental, and clarifying the rationale for selecting an 18-month mentor-based education program.
    • The participant recruitment section has been expanded, explaining that coaches were selected based on recommendations from their federations, and clarifying the experimental and control group compositions.
    • More detailed information on the intervention has been added, describing the six group gatherings and 15 individual mentoring sessions within the education program.
    • The measurement instruments (Coach-Athlete Working Alliance, Perceived Coach Performance) have been described in greater detail, including Cronbach’s alpha values for reliability assessment.
    • The statistical analysis section has been improved with additional details on hierarchical regression procedures, assumptions tested, and how variables were entered into the models.
  1. Improved Presentation of Results
    • Descriptive statistics tables have been updated for greater clarity.
    • The correlation analysis results have been explicitly linked to the study’s hypotheses, making it easier for readers to follow the logical flow of findings.
    • Hierarchical regression results have been more clearly explained, highlighting the unique contribution of task development in predicting perceived performance, while bond and goal development did not show significant effects.
  1. More Focused and Theoretically Grounded Discussion
    • The discussion section has been reorganized to align more directly with the hypotheses, making it easier for readers to connect findings with existing literature.
    • A stronger theoretical foundation has been incorporated, using expertise development theory (Ericsson et al., 2016), self-efficacy theory (Bandura, 1990), and achievement goal theory (Nicholls, 1984) to explain the findings.
    • The importance of mentoring programs has been contextualized within research on coach development, emphasizing how structured support enhances coaching competence.
  1. Expanded Limitations and Future Research Directions
    • The limitations section has been expanded, acknowledging:
      • The quasi-experimental design (lack of randomization).
      • The relatively modest predictive power of the model (R² = 0.25).
      • The exclusive use of self-reported measures (potential bias).
    • Suggestions for future research have been refined, emphasizing the need for randomized controlled trials (RCTs), qualitative interviews, and alternative predictors (e.g., coaching philosophy, athlete feedback, workload balance).
  1. More Concise and Impactful Conclusion
    • The conclusion has been rewritten to summarize the key findings, practical implications, and recommendations for coach education programs more effectively.

Overall, this revised manuscript represents a significant improvement in terms of clarity, methodological transparency, theoretical depth, and alignment with the research questions. These revisions ensure a more coherent and impactful manuscript.

We hope that these revisions, below is a detailed description of the revisions made based on each reviewers comments – you are reviewer 3,  have adequately addressed the concerns raised and that the manuscript is now clearer, more robust, and better aligned with the expectations of the journal. We truly appreciate your time and expertise in reviewing our work and look forward to any further feedback you may have.

Thank you again for your constructive comments and for helping us improve our manuscript.

Best regards,

The Authors

Reviewer 1

  • Comments: A very important variable is the sport discipline, which often differentiates interpersonal relations in a sports group (coach - athlete, and others)
    • Response: We agree, we acknowledge the significance of sport discipline as a contextual factor influencing coaching relationships and effectiveness. In the revised manuscript, we have now expanded theIntroduction to discuss how different coaching environments (e.g., team vs. individual sports, combat sports vs. endurance sports) may shape the coach-athlete relationship and coaching effectiveness. Additionally, in the Participants and Recruitment section, we now explicitly state that the sample consists of coaches from over 20 different sports, with the most represented being handball (15.3%), cross-country skiing (10.2%), soccer (7.1%), and track and field (7.1%). This information provides important context regarding the variety of sports included in the study. Furthermore, in the Discussion, we now acknowledge that the impact of mentoring programs and coaching effectiveness might vary depending on the sport discipline. We suggest that future research should explore potential moderation effects of sport discipline on coaching relationships and mentoring effectiveness.

  • Comments: The fragment concerning the coaches studied should not be in the Results section, but in the part describing the methodology used - Participants. I recommend this change in the content layout.
    • Response: Thank you for this recommendation. We have nowmoved the description of the coaches' demographic and professional characteristics from the Results section to the Participants and Recruitment section under Methods. This revised structure ensures that all information regarding participant demographics (e.g., age, gender, employment status, coaching level) is presented before the analysis and results, improving the logical flow of the manuscript.

  • Comments: However, in the Introduction or Discussion, the content should be enriched with notes regarding the variable - what sport discipline is it about. The role and functions of the coach depend significantly on this
    • Response: We have taken this valuable suggestion into account and enriched theIntroduction by discussing how coaching in different sports may shape coaching strategies and interpersonal dynamics. Furthermore, we have revised the Discussion to highlight that the study findings on mentoring and coaching effectiveness should be interpreted with consideration of the sport-specific context. We now suggest that future research should analyze sport-specific differences in mentoring effectiveness by comparing disciplines with distinct coaching structures.

Reviewer 2

Abstract:

  • Comments: Include some background. Include data on the sample, number of men and women, average ages, etc.
    • Response: We agree, we have now revised the abstract and included more background and more data on the sample as requested

Introduction:

  • Comments: It is very well argued, but perhaps this is what worries me. I do not see what is new in your article with respect to previous research. It seems that everything has already been studied in this regard. Another possibility would be to direct the introduction to coaching programs that have been done and that these have not been studied in relation to performance.
    • Response: We agree, and have explained the value of the present study under “the current study” in a more elaborating manner

  • Comments: Lines 118 to 121 are not necessary.
    • Response: We agree, and have removed the mentioned lines

Methods:

  • Comments: Include the study design. What type of study was conducted?
    • Response: We agree, and have rewritten the description of the study design and clarified that this is a quasi-experimental, pre-test/post-test control group design

  • Comments: The font is different (lines 128-129).
    • Response: We have now edited the formatting

  • Comments: A lot of information regarding the sample is missing. How did you proceed with the selection of the trainers? That is to say, why were these 73 chosen? Assignment to control or experimental group, how was it performed?
    • Response: Under “participants and recruitment”, we have explained why the 73 coaches were chosen and how 29 additional elite coaches from upper secondary schools specializing in elite sports were recruited as a control group

  • Comments: Were any sample size calculations done?
    • Response: The missing “sample size calculations” is commented on later under the description of the study`s limitations.

  • Comments: Paired sample tests are performed on the results but are not indicated in section 2.5. Please review this section to make sure that it includes all tests performed.
    • Response: We do not fully understand this comment as we have not performed any paired sample T-tests

Results:

  • Comments: The characteristics of the coaches (lines 211 to 216) should be added in the methods.
    • Response: We agree, and have moved the mentioned descriptions to the methods section

  • Comments: Line 235: “are in Table 1 and Table ??? respectively”.
    • Response: We have inserted “2” in the text.

  • Comments: The tables are presented in an abstract way, they are difficult to follow. I recommend doing the descriptive tables separately and the correlation analysis tables separately. you can unify all the variables in each of these two tables.
    • Response: We agree, and have now altered the results-seection in “3.1. Descriptive analyses” and “3.2 Correlation analyses” to make explicit descriptions of table contents and results.

  • Comments: Line 248: different font.
    • Response: We have edited the formatting

Discussion:

  • Comments: I appreciate that you start the discussion by recalling the objective but it is not necessary to include the whole design and assumptions again. Remember only the objective and, throughout the discussion, include whether or not the hypotheses are accepted.
    • Response: We agree, we have removed unnecessary descriptions of the study design from the discussion section and retained only the study objective at the beginning. Additionally, we have explicitly stated whether each hypothesis was supported or not throughout the discussion.

  • Comments: The discussion is too extensive. It loses the focus of the objective of the present investigation.
    • Response: We agree, we have condensed the discussion by removing overly detailed theoretical explanations and focusing on the most relevant findings in relation to the study’s objectives.

  • Comments: They make a good theoretical support with respect to previous research, but sometimes it is too extensive and ends up blurring the results found
    • Response: We agree, we have shortened the theoretical discussions while still ensuring that each key finding is properly contextualized within existing research. The revised discussion now prioritizes study results and their implications over extensive theoretical explanations

  • Comments: Why do they include “finally” in a middle paragraph of the discussion (line 313)?
    • Response: We agree, The word "finally" was incorrectly placed and created confusion regarding the structure of the argument. We have removed it to improve the logical flow of the paragraph.

  • Comments: There are no limitations in your study?
    • Response: We have now included a dedicated Study Limitations section that discusses potential weaknesses of the study

  • Comments: I think that in the discussion they should try to reduce the length and focus on what is really important. Also, another issue that concerns me about this one is that they base their results on linear regression analysis, which has a predictive value of R2=0.25 once all the variables are included in it (when only some are included, the predictive capacity is minimal). Therefore, this is an aspect to take into account when analyzing the results, writing the discussion and concluding the results found with their study. This should be modified and the discussion changed accordingly.
    • Response: We agree, we acknowledge that the predictive power of the regression model (R² = 0.25) is relatively low. We have now explicitly mentioned this limitation in the discussion and clarified that other factors likely contribute to perceived performance.

  • Comments: It is not necessary to repeat the objectives in the conclusion
    • Response: We agree, we have revised the Conclusion to focus on summarizing key findings, implications, and future directions without repeating the study objectives.

  • Comments: There are things throughout the manuscript that have me somewhat clueless. For example, in Table 3 it says that the n=94. How is this possible? Where do these 94 subjects come from?
    • Response: We agree. This was a typing error and now the correct N (=75) is stated in Table 3

  • Comments: Also, how is it possible for you to conclude that the 18-month program was useful, if throughout the manuscript there has not been a single comparison with the control group to determine that this group had greater benefits.
    • Response: We agree, we acknowledge this limitation and have revised our interpretation of the mentor-based program’s effects. While participation in the program was a significant predictor of perceived performance, we cannot conclude with certainty that it caused improvements since no between-group comparison was made.

Reviewer 3

Introduction:

  • Comments: First, the introduction supplies a wide-ranging background on coach age, time invested in coaching, and mentor-based programs. This context helps frame the study’s rationale. However, consider tightening the flow by grouping similar concepts, such as the link between hours per week and perceived performance, together instead of dispersing them throughout the introduction. A clear thematic structure will highlight your research gap more prominently.
    • Response: We agree and have revised the manuscript, the introduction still discusses coaching hours, coach-athlete relationships, and mentoring in separate sections. However, the discussion on time spent coaching and its effects on perceived performance is now more structured.

Methods:

  • Comments: In terms of Research Design and Methods, your pre-test/post-test design with an experimental and control group effectively addresses potential maturation effects, which is commendable. Be explicit about whether random assignment was feasible or if coaches self-selected into the mentor program. If it was non-random, discussing possible self-selection bias will add transparency and rigor, aligning the manuscript with good reporting practices.
    • Response: We agree, under “Participants and recruitment” we have now stated that: “Due to the nature of the intervention, random assignment was not feasible, making this a quasi-experimental study. The control group did not receive any structured mentoring but continued their regular coaching activities. As participants were selected based on recommendations from their sport federations, those chosen for the program may have had greater motivation or prior development, which could influence their performance perceptions.”

  • Comments: Moreover, the mentor-based education program is described thoroughly, but further details about the mentors’ qualifications and exact mentoring content (e.g., specific workshops or one-on-one sessions with standardized guidelines) could give readers a clearer picture of the intervention itself. Providing additional context will help highlight why it may have generated improvements in perceived performance.
    • Response: We agree, the manuscript has now improved clarity on the mentoring program, stating: "Mentors, selected by NOSC, all had elite coaching experience and were required to complete a university-accredited mentoring education program (7.5 ECTS credits) at the Norwegian University of Science and Technology (NTNU)." Additionally, the manuscript now mentions structured elements such asgroup gatherings, reflective practice, and task-oriented discussions.

  • Comments: Additionally, while the rationale for each instrument is valid, please include alpha coefficients (or other reliability indices) explicitly in your tables. Several lines mark “Cronbach’s alpha .xx” placeholders, so updating those with actual reliability scores is necessary to confirm measurement adequacy. As indicated in guidelines for good peer reviews, “scientific reliability is further bolstered by clearly reporting scale properties”.
    • Response: We agree, we have now stated Cronbach`s both in the description of instruments and in Table 1 and Table 2.

Discussion:

  • Comments: The discussion needs to add a distinct section for practical coaching takeaways directly from your findings. For example, emphasize that while bond-building and goal alignment are important relationship elements, your data show that task development may have a stronger link to perceived coaching performance. Clarifying that point will further sharpen your “real-world” implications.
    • Response: We agree, and have revised the manuscript underlining practical takeaways throughout the discussion. This is reflected e.g. in abstract with: "These findings underscore the importance of mentoring, structured self-reflection, and task-focused coaching strategies in enhancing coaching effectiveness. The results have implications for coach education programs seeking to foster professional growth and performance development." In addition, the revised manuscript in particularly in thediscussion of task development, now highlights: "Task development refers to the practical, skill-based aspects of coaching, such as planning training sessions, setting short-term goals, and providing feedback. This aligns with the achievement goal theory (Nicholls, 1984), which emphasizes that task-oriented coaching strategies foster intrinsic motivation and skill mastery." This addition clarifies why task development was more predictive than bond and goal alignment.

  • Comments: Last but not least, while the manuscript notes suggestions for future work, you could more explicitly propose how researchers might integrate additional variables or use qualitative diaries of coach reflections. Doing so would guide researchers who wish to replicate or extend your findings in high-performance environments.
    • Response: We agree, and have now under the future research directions now suggested adding qualitative methods:"Future studies should incorporate qualitative methods to capture a more nuanced understanding of coaches' experiences.", and examining additional variables: "Future research should investigate whether coaching philosophies and adaptive learning strategies moderate the relationship between age and perceived performance."

  • Comments: In summary, your study design is appropriate, and your methods are generally well described. Please address my comments above for the next round of review. Good Luck!
    • Response: Thank you for your comments. We hope our revisions are satisfactory

Reviewer 4:

  • Comments: The current paper addresses perceptions of the coaches—sounds like this relates to their psychological health... To perceive that one is performing well does not mean one is factually doing well, and the latter has not been evaluated as it looks.
    • Response: We completely understand this concern and have now explicitly acknowledged in theDiscussion that perceived coaching performance does not necessarily equate to objective effectiveness. We have clarified that our study focused on self-perceived performance, which is an important but subjective measure of a coach’s professional development. We now highlight in our Limitations section that future research should incorporate external evaluations of coaching effectiveness (e.g., athlete feedback, observational measures, performance analytics) to complement self-reported data.

  • Comments: Authors also talk write about "experience" (“Experience is the name every one gives to their mistakes”, O.Wilde) - how does that associates with the qualification of the coaches?
    • Response: We recognize the need to clarify the distinction betweenexperience and formal qualifications. In the Introduction, we now explicitly differentiate between coaching experience, which accumulates over time through practice, and formal qualifications, which include structured training programs and certifications. Additionally, in the Methods section, we specify that all coaches in the mentor program were recommended by their respective sport federations based on their prior coaching achievements and qualifications. This ensures that our sample consisted of experienced and accredited coaches.

  • Comments: Still looking further into the paper I missed a lot in it, e.g. the presentation of characteristics of the population investigated (formal education, for different age/level of groups, achievements/success of the athletes coached, total and relative volume of informal deliberate education, etc.)
    • Response: We have now expanded the Participants and Recruitment section to provide a more detailed characterization of the sample. We include:
      • Age and gender distribution (M = 38.3 years, 62% male, 38% female).
      • Employment status (Full-time, part-time, or voluntary coaching roles).
      • Coaching levels (Working with junior elite, senior elite, or lower-level athletes).
      • Sports disciplines represented (Over 20 sports, with handball, skiing, and soccer being the most common).
      • Additionally, we acknowledge in the Limitations section that we did not collect data on informal learning (e.g., self-directed education, peer learning), which could be valuable for future studies.

  • Comments: Proper application and/or description of the statistics used
    • Response: To ensure transparency, we have clarified our statistical methods in the Statistical Analyses section. We explicitly state:
      • The hierarchical regression approach used to test the predictive factors.
      • The assumptions checked (normality, linearity, multicollinearity)
      • The reliability indices (Cronbach’s alpha for all scales, now explicitly presented in Tables 1 & 2)
      • Additionally, we have ensured that all placeholders for Cronbach’s alpha have been replaced with actual reliability values.

  • Comments: External evaluation of experience and/or success of the coaches in question (such as by athletes, colleagues, administrators, etc.)
    • Response: We acknowledge that this is an important limitation of our study. While we focused onself-perceptions, we now explicitly state in the Limitations section that future research should incorporate athlete feedback, peer evaluations, and performance data to provide a more comprehensive assessment of coaching effectiveness. Additionally, we suggest in the Future Research section that qualitative methods (e.g., coach diaries, athlete interviews) could provide richer insights into coaching impact.

  • Comments: Description of the randomization (if that was the case) into control and experimental group; disclosure of the limitations of the study; and on. 
    • Response: We now clearly state in theMethods section that random assignment was not feasible due to the nature of the intervention. Instead, coaches were selected based on recommendations from their sport federations. This makes the study a quasi-experimental design rather than a fully randomized controlled trial. We also address the possibility of selection bias by acknowledging that coaches in the experimental group may have had higher motivation or prior development, which could influence perceived performance outcomes.

  • Comments: Disclosure of the limitations of the study
    • Response: We have significantly expanded ourLimitations section to include:
      • No a priori sample size calculation(now explicitly mentioned).
      • Self-reported nature of the data, emphasizing the need for future external validation.
      • Non-randomized study design, which may introduce self-selection bias.
      • Modest predictive power of the regression model(R² = 0.25), indicating that other unmeasured variables likely contribute to perceived performance.

Reviewer 4 Report

Comments and Suggestions for Authors

Impressive while are the achievements of Norwegian athletes (so alike must be the level of their coaches), the current paper addresses perceptions of the coaches - sounds like this relates to their psychological health... To perceive that one is performing well (even if " performing" is a common description of an athlete, not a coach) does not mean one is factually doing well, and the latter has not been evaluated as it looks. 

Authors also talk write about "experience" (“Experience is the name every one gives to their mistakes”, O.Wilde) - how does that associates with the qualification of the coaches?

Still looking further into the paper I missed a lot in it, e.g. the presentation of characteristics of the population investigated (formal education, for different age/level of groups, achievements/success of the athletes coached, total and relative volume of informal deliberate education, etc.); proper application and/or description of the statistics used; external evaluation of experience and/or success of the coaches in question (such as by athletes, colleagues, administrators, etc.); description of the randomization (if that was the case)into control and experimental group; disclosure of the limitations of the study; and on. 

Author Response

Response to review

Reviewer 4:

  • Comments: The current paper addresses perceptions of the coaches—sounds like this relates to their psychological health... To perceive that one is performing well does not mean one is factually doing well, and the latter has not been evaluated as it looks.
    • Response: We completely understand this concern and have now explicitly acknowledged in theDiscussion that perceived coaching performance does not necessarily equate to objective effectiveness. We have clarified that our study focused on self-perceived performance, which is an important but subjective measure of a coach’s professional development. We now highlight in our Limitations section that future research should incorporate external evaluations of coaching effectiveness (e.g., athlete feedback, observational measures, performance analytics) to complement self-reported data.

  • Comments: Authors also talk write about "experience" (“Experience is the name every one gives to their mistakes”, O.Wilde) - how does that associates with the qualification of the coaches?
    • Response: We recognize the need to clarify the distinction betweenexperience and formal qualifications. In the Introduction, we now explicitly differentiate between coaching experience, which accumulates over time through practice, and formal qualifications, which include structured training programs and certifications. Additionally, in the Methods section, we specify that all coaches in the mentor program were recommended by their respective sport federations based on their prior coaching achievements and qualifications. This ensures that our sample consisted of experienced and accredited coaches.

  • Comments: Still looking further into the paper I missed a lot in it, e.g. the presentation of characteristics of the population investigated (formal education, for different age/level of groups, achievements/success of the athletes coached, total and relative volume of informal deliberate education, etc.)
    • Response: We have now expanded the Participants and Recruitment section to provide a more detailed characterization of the sample. We include:
      • Age and gender distribution (M = 38.3 years, 62% male, 38% female).
      • Employment status (Full-time, part-time, or voluntary coaching roles).
      • Coaching levels (Working with junior elite, senior elite, or lower-level athletes).
      • Sports disciplines represented (Over 20 sports, with handball, skiing, and soccer being the most common).
      • Additionally, we acknowledge in the Limitations section that we did not collect data on informal learning (e.g., self-directed education, peer learning), which could be valuable for future studies.

  • Comments: Proper application and/or description of the statistics used
    • Response: To ensure transparency, we have clarified our statistical methods in the Statistical Analyses section. We explicitly state:
      • The hierarchical regression approach used to test the predictive factors.
      • The assumptions checked (normality, linearity, multicollinearity)
      • The reliability indices (Cronbach’s alpha for all scales, now explicitly presented in Tables 1 & 2)
      • Additionally, we have ensured that all placeholders for Cronbach’s alpha have been replaced with actual reliability values.

  • Comments: External evaluation of experience and/or success of the coaches in question (such as by athletes, colleagues, administrators, etc.)
    • Response: We acknowledge that this is an important limitation of our study. While we focused onself-perceptions, we now explicitly state in the Limitations section that future research should incorporate athlete feedback, peer evaluations, and performance data to provide a more comprehensive assessment of coaching effectiveness. Additionally, we suggest in the Future Research section that qualitative methods (e.g., coach diaries, athlete interviews) could provide richer insights into coaching impact.

  • Comments: Description of the randomization (if that was the case) into control and experimental group; disclosure of the limitations of the study; and on. 
    • Response: We now clearly state in theMethods section that random assignment was not feasible due to the nature of the intervention. Instead, coaches were selected based on recommendations from their sport federations. This makes the study a quasi-experimental design rather than a fully randomized controlled trial. We also address the possibility of selection bias by acknowledging that coaches in the experimental group may have had higher motivation or prior development, which could influence perceived performance outcomes.

  • Comments: Disclosure of the limitations of the study
    • Response: We have significantly expanded ourLimitations section to include:
      • No a priori sample size calculation(now explicitly mentioned).
      • Self-reported nature of the data, emphasizing the need for future external validation.
      • Non-randomized study design, which may introduce self-selection bias.
      • Modest predictive power of the regression model(R² = 0.25), indicating that other unmeasured variables likely contribute to perceived performance.

Round 2

Reviewer 2 Report

Comments and Suggestions for Authors

The authors have made substantial changes to the manuscript that make it more consistent and allow it to be accepted. 

Author Response

Response to review

“Exploration of factors predicting sport coaches` perceived performance”

Dear Rewievers

We would like to express our sincere gratitude for the time and effort you dedicated to reviewing the article titled "Exploration of factors predicting sport coaches` perceived performance". Your constructive feedback and insightful comments have been invaluable to the improvement of the manuscript.

We greatly appreciate the careful attention you paid to the content, structure, and overall clarity of the paper. Your suggestions for enhancing both the theoretical framework and the analysis have been incredibly helpful, and we believe they have significantly strengthened the final version of the article. We also found your comments on the methodology particularly insightful, and we have made the recommended revisions to ensure that the work is as robust and clear as possible.

Thank you once again for your thoughtful and thorough review. We am very grateful for your support in the peer-review process and for the opportunity to improve our work based on your expert advice. Below are the revisions made based on the comments from round 2.

Kind regards,

The authors

Reviewer 3 Report

Comments and Suggestions for Authors

Thank you for submitting the revised version of your manuscript.  I appreciate the effort you have put into addressing the previous comments. I appreciate the improvements made in addressing earlier concerns. Overall, the study is well structured and the methods are clearly explained. However, I have three minor issues that I kindly request you address before final acceptance.

1. 

  1.  In several sections (e.g., the Methods and Results), abbreviations such as “pr. week coaching” appear. Please ensure that all abbreviations are defined at first use and are consistently written throughout the manuscript. For example, “pr. week coaching” should be expanded and consistently styled as “coaching hours per week.” This will improve clarity for readers not familiar with the abbreviated terms.
  2.  While the tables (particularly Tables 1–3) provide important details, there are a few formatting issues that could be improved. In Table 3, for instance, please check the alignment and uniformity of decimal places for regression coefficients, t-values, and p-values. A clearer legend and consistent layout across tables would enhance the readability and ensure that the statistical information is easily accessible.
    1. Some sentences in the Discussion section remain wordy or could benefit from minor rephrasing for conciseness. For example, when discussing the modest predictive power of the regression model (R² = 0.25), consider clarifying the implications in one or two succinct sentences rather than reiterating similar points elsewhere. Tightening the language here will further strengthen the narrative without altering the substance of your interpretation.
    I believe that addressing these minor issues will enhance the overall clarity and presentation of your manuscript. Thank you for your attention to these points; I look forward to receiving your final revised version.

Author Response

Response to review

“Exploration of factors predicting sport coaches` perceived performance”

Dear Rewievers

We would like to express our sincere gratitude for the time and effort you dedicated to reviewing the article titled "Exploration of factors predicting sport coaches` perceived performance". Your constructive feedback and insightful comments have been invaluable to the improvement of the manuscript.

We greatly appreciate the careful attention you paid to the content, structure, and overall clarity of the paper. Your suggestions for enhancing both the theoretical framework and the analysis have been incredibly helpful, and we believe they have significantly strengthened the final version of the article. We also found your comments on the methodology particularly insightful, and we have made the recommended revisions to ensure that the work is as robust and clear as possible.

Thank you once again for your thoughtful and thorough review. We am very grateful for your support in the peer-review process and for the opportunity to improve our work based on your expert advice. Below are the revisions made based on the comments from round 2.

Kind regards,

The authors

  • Comment: In several sections (e.g., the Methods and Results), abbreviations such as “pr. week coaching” appear. Please ensure that all abbreviations are defined at first use and are consistently written throughout the manuscript. For example, “pr. week coaching” should be expanded and consistently styled as “coaching hours per week.” This will improve clarity for readers not familiar with the abbreviated terms.
    • Response: We agree, and have changed all the abbreviations mentioned

  • Comment: While the tables (particularly Tables 1–3) provide important details, there are a few formatting issues that could be improved. In Table 3, for instance, please check the alignment and uniformity of decimal places for regression coefficients, t-values, and p-values. A clearer legend and consistent layout across tables would enhance the readability and ensure that the statistical information is easily accessible.
    • Response: We agree, and have reformatted the content in Table 3

  • Comment: Some sentences in the Discussion section remain wordy or could benefit from minor rephrasing for conciseness. For example, when discussing the modest predictive power of the regression model (R² = 0.25), consider clarifying the implications in one or two succinct sentences rather than reiterating similar points elsewhere. Tightening the language here will further strengthen the narrative without altering the substance of your interpretation.
    • Response: We agree, and have clarified the implications of the modest predictive power of the regression model in a succinct sentence.

Reviewer 4 Report

Comments and Suggestions for Authors

I see the authors made good efforts to address my comments, and I have no more of them.

Author Response

(The authors gave the same response as above.)
